# Land Use, Microorganisms, and Soil Organic Carbon: Putting the Pieces Together

**Lingzi Mo** [1,2], **Augusto Zanella** [2,*], **Cristian Bolzonella** [2], **Andrea Squartini** [3], **Guo-Liang Xu** [1], **Damien Banas** [4], **Mauro Rosatti** [5], **Enrico Longo** [5], **Massimo Pindo** [6], **Giuseppe Concheri** [3], **Ines Fritz** [7], **Giulia Ranzani** [2], **Marco Bellonzi** [2], **Marco Campagnolo** [8], **Daniele Casarotto** [2], **Michele Longo** [2], **Vitalyi Linnyk** [2], **Lucas Ihlein** [9] and **Allan James Yeomans** [10]

1    School of Geography and Remote Sensing, Guangzhou University, Guangzhou 510006, China
2    Department Land Environment Agriculture and Forestry, University of Padua, Viale dell'Università 16, 35020 Legnaro, Italy
3    Department Agronomy, Food, Natural Resources, Animals, Environment, University of Padua, Viale dell'Università 16, 35020 Legnaro, Italy
4    UR AFPA—INRA, Université de Lorraine, Faculté des Sciences, Boulevard des Aiguillettes, BP 70239, 54500 Vandœuvre-lès-Nancy, France
5    Albarella s.r.l., 45010 Rosolina, Italy
6    Fondazione Edmund Mach, 38098 San Michele all'Adige, Italy
7    Department IFA-Tulln, University of Natural Resources and Life Sciences, Konrad Lorenz Str. 20, 3430 Tulln, Austria
8    Veneto Agricoltura, Viale dell'Università 14, 35020 Legnaro, Italy
9    Faculty of Arts, Social Sciences and Humanities, University of Wollongong, Wollongong, NSW 2522, Australia
10    Yeomans Plow Company, Gold Coast, QLD 4217, Australia
*    Correspondence: augusto.zanella@unipd.it; Tel.: +33-78-3615-957

**Abstract:** We set out to study what biodiversity is, and how it can be influenced by human activities. To carry out this research, we looked for two, relatively closed, natural small-island systems: one little-influenced by human settlement and another equivalent (same vegetation series aligned 200 m from the first) but heavily settled. In these two environments, two transects were created in 10 subecosystems, from the sea to the mainland. We sought similar subecosystems in both places. We selected a series of eight points along the same gradient in the two environments, with two additional nonoverlapping points, specific to "natural plus" or "natural minus". We studied soil microorganisms and arthropods to have a large number of cases (OTUs) available, and also studied the microorganisms' phylogenetic status. We also compared biodiversity with soil organic carbon (SOC) content, using two SOC measurement systems (with and without litter), to understand biodiversity starting from its potential source of food (SOC). The results surprised us: the biodiversity indices are higher in the anthropized environment; the level of biodiversity of these microorganisms (OTUs) is linked to the quantity of organic carbon measured in the first 30 cm of soil with two different methods, Carbon Still Yeomans (650 g of soil sample) and Skalar Primacs ATC-100-IC-E (1 g of soil sample). The following forced line at the origin explains 85% of the variance: Shannon–Wiener's H = 1.42 • ln (TOC400); where ln = natural logarithm and TOC400 = organic carbon lost from a soil sample raised to 400 °C. The concept of biodiversity merges with that of survival: the more species there are, the better they are organized among themselves in the process of food consumption (SOC utilization), and the better they will be able to transform the environment to survive and evolve with it. We wanted to identify the differences in soil biodiversity of natural and anthropogenic ecosystems, to offer evidence-providing tools to land managers to achieve more ecologically efficient managing practices.

**Keywords:** soil microorganisms; soil biodiversity; anthropized vs. natural; phylogenetic turnover; SOC; Carbon Still; Skalar Primacs; TOC400

## 1. Introduction

The article we present is the result of four fundamental elements. (1) In the underlying plot, there is a question that scientific thought has always tried to answer: "what is the biodiversity of planet Earth and on what is it based?" (2) More accessible on the surface is instead another question: "in the evolution of microbial biodiversity, what important biological role are humans playing?" (3) Could it be possible to find a mathematical formula that links soil organic carbon (SOC) storage and soil microbial biodiversity? To measure SOC, we used two ignition weight-loss methods, one working on 1 g and one on 600 g of sample. (4) The whole work revolves around a research program financed by the University of Padua and a private association, which jointly attempt to operate a sustainable economic and social exploitation of a Mediterranean island.

1. It is not easy to circumscribe the concept of biodiversity. It is closely linked to those of life and evolution: (1) to that of life, because we think that to live means "to complexify", (a) in the sense of Lynn Margulis [1], to become more and more complex by symbiosis with other living beings; or (b) even to increase at ever larger scales, from cell to individual, from ecosystem to the entire planet or universe, in the spirit of Gaia [2]; (2) to that of evolution, on the scale of the individual in the population of a species, to conform to a changing niche [3]. In this context, the definition of biodiversity should be considered in parallel with those of complex organism or biotic community [4], climax [5], and ecosystem [6]. In such a confrontation of concepts, biodiversity acquires a finality, as it was a living and growing complex entity. Tansley accepted the concepts of climax and biome, but decisively rejected the similar ones of complex organism and biotic community. His judgment is accepted as a definitive choice by the scientific community [7,8] and is generally considered to be a fundamental law for modern ecology. However, it still divides at least the authors of this article, who are of different nationalities and representatives of a wide range of human activities, such as research, economic and ecological management, manufacturing, commerce, art and teaching, and some students in forest sciences. A fundamental fact that depends on this comparison of concepts has remained unresolved: on planet Earth, since its abiotic origin and up to the present day, despite well-known periods of crisis, biodiversity continued to increase [9,10]. Is the fact of growing, on average (and not linearly), a fundamental law, or a consequence linked to abiotic factors? Let us not forget that life on the planet started from a very inhospitable environment [11], adapting it to its needs [2]. The answer to this question of intrinsic growth determines how to sustain biodiversity in the coming years and in a changing climate. The way out seems to inevitably involve soil living storage power and management [12–19].

2. Ample theoretical and empirical work has shown that interactions of human activities with the ecosystems are dynamic and complex [20,21]. The rapidly increasing human population and the increasing ecological footprint per capita are placing further pressure on the natural landscapes [22]. Despite a growing awareness of the risks at which humans and other species are, and will be exposed as a result of anthropogenic activities, pressure continues to grow, whether to meet the vital needs of a positive demography or for leisure activities such as tourism, whose environmental footprint is also growing [13]. Biodiversity on the Earth is indeed highly affected by anthropic activities fundamentally causing alterations to the environment [23]. Biodiversity is known as a critical determinant of ecosystem functioning, starting from the ground level, i.e., the soil itself [24]. Therefore, understanding biodiversity changes is an important issue that promotes better knowledge and finer capability to estimate how soil biodiversity changes might impact ecosystem sustainability [25]. However, previous studies of anthropic effects on ecosystems have overwhelmingly focused on macroscopic plants and animals [22], both on land and in water. On a local scale, the microbial communities are more influenced by soil properties and microclimate changes than plant communities [26,27]. At the microscale, the effects on soil biodiversity are far less known, despite the importance of soil organisms (bacteria, fungi,

arthropods, invertebrates, etc.), which are the major regulators of essential ecosystem functions and services, such as plant productivity, nutrient cycling, organic matter decomposition, and pollutant degradation [28–33]. One of the important human activities, tourism, accounting for about 8% of global greenhouse gas emissions [13], is continuing to increase. At more local scales, the touristic–recreational use of territories may endanger their environmental value and that of the surrounding areas. An increase in the popularity of outdoor leisure activities leads to an increase in the number of visitors in the same tourist area. This increase may significantly affect belowground ecosystems, especially microbial communities [34]. Lucas-Borja et al. (2011) studied the microbiological properties of soil and vegetation in Mediterranean mountains, and the result indicated that increased tourist activity significantly impacted soil microbial processes and vegetal communities, mainly due to soil compaction [35]. A study in Finland also showed that continuous human trampling causes significant changes in soil microbial functions, even with light stepping [36]. Understanding the extent of such anthropic pressure can provide insights informing future environment management, restoration, and monitoring. In recent years, methods for studying microbial communities have progressed rapidly, and DNA sequencing has allowed a more thorough understanding of the key members of soil microbial communities and biodiversity patterns [37,38]. Changes in soil physical and chemical properties have been reported to shape the composition of a microbial community, in terms of density, diversity, and activity [39–44]. Moreover, soil microbes have a fundamental role in soil responses to human disturbances since they are tightly dependent on the surrounding abiotic and biotic environment [45–47]. The differences in physiology and ecology of bacteria and fungi suggest that they would be controlled by different environmental factors [48]. Previous studies have shown that fungi may be more sensitive than bacteria to changes in vegetation [49], and shifts in carbon pools may have different effects on bacteria and fungi [50,51]. The importance of understanding community assembly processes is broadly recognized in microbial ecology [52,53], and the assembly of microbial communities is known to be influenced by both deterministic and stochastic processes [54,55]. Deterministic processes refer to habitat filtering or biotic interactions such as mutualism, commensalism, and parasitism, while stochastic processes refer to random demographic changes in mortality and passive dispersal [53,56,57]. Wang et al. (2019) pointed out that the mechanisms of soil bacterial and fungal community assembly are different [58]. Thus, elucidating the differential dynamics and factors that affect microbial community structuring can help in estimating how anthropogenic environmental changes ultimately impact the different types of microbial communities, which represent a vast and still largely obscure component of planetary biodiversity.

3. To work on SOC and biodiversity, we capitalized on the occurrence of a naturally preserved peninsular area, just adjacent to a symmetrically distributed strip of land (see aerial view in Figure 1) that had the same original soils and vegetation, but that had been, for the last forty years, fully transformed into a tourist resort. The availability of these two neighboring sites—one of which is now exploited by human recreational business, and the other preserved by conservation law—allowed us to investigate the effects of human-driven ecosystem manipulation, in comparison to a more natural landscape evolution. Of course, it is important to note that there is no longer anything "natural" (in the sense intended by Clements, 1936; or Tansley, 1935) on our planet today. It is more correct to assume that the whole environment is now, to varying degrees "human-influenced". Every place on the planet can be situated on a spectrum ranging from the completely urbanized, to what remains of areas that could be defined as "moderately altered by human action", to those very rare that still appear as "wilderness". Even so-called wilderness areas are increasingly affected by global atmospheric climatic and pollution drifts, meaning that they are far from pristine [59]. Here at the two sites, one more-natural and one more-man-made, we

studied 10 ecosystems along a gradient of increasing complexity from the poorest in species near the shore to the rich and wooded hinterland. Referring to a concept of soil as the place where each ecosystem starts its formation and evolution [60–63], we delved into the organic resources available to these living beings, trying to link the total organic carbon of the soil, as if it were a source of nourishment, to its biodiversity.

4. Islands are, moreover, ecologically isolated landmasses, further qualifying as useful model systems to address ecological questions (Warren in [64]). Here, we aimed at addressing the following questions: (i) How does soil microbial diversity change under human pressure such as habitat fragmentation and/or recreational tourism exploitation? (ii) What are the impacts of different land management practices on soil biodiversity? (iii) Are assembly mechanisms of soil microbial communities different in natural and anthropogenic ecosystems? (iv) Can ecological diversity indices be suitable to answer these questions? Finally, (v) is it possible to better define the concept of biodiversity itself?

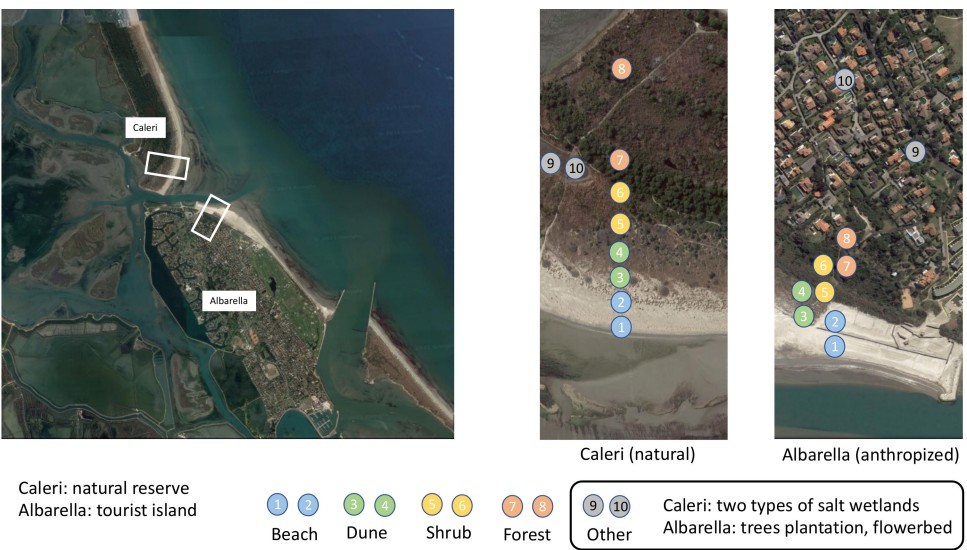

**Figure 1.** Sampling panel. The leftmost panel shows the symmetrical location and common geological and vegetational origin of the two opposite sites, Porto Caleri the natural reserve, versus Albarella, the touristic island. In both sites (center and rightmost panels), two series of samples were compared, belonging to five ecosystems (beach, dune, shrub, forest, and other).

## 2. Materials and Methods

### 2.1. Site Description and Sampling

The study was conducted at two peninsular strips of land within the Po River delta lagoon system, located in the southern part of Veneto, Italy. Both sites are physically separated from land by a series of channels and larger water bodies bridged to reach the sites (Figure 1). The island of Albarella (45°04′32″ N 12°20′38″ E) features many constructions, mostly visitors' accommodations, leisure facilities, and economic activities (it receives about 110,000 visitors a year, with an average stay of 10 days on the island). Across a channel of the sea, less than a kilometer on its northern side, it is faced by Porto Caleri (45°05′53″ N 12°19′27″ E), an area under governmental protection and restricted access since 1990, where only walking within fenced wooden trails is permitted, and is defined as 'Giardino Botanico Litoraneo del Veneto' ("Coastal Botanical Garden of Veneto"), representing a preserved, nearly natural ecosystem. Visits are devoted to educational purposes serving to illustrate the natural vegetation transect of all the local coast habitats from forest to seashore. In this investigation, we focused on (1) a full set of vegetational stages in both sites, from (i) beach (2 phytocoenosises) to (ii) dune (2 phytocoenosises), (iii) shrub (2 phytocoenosises), and (iv) forest (2 phytocoenosises); and (2) on two peculiar biotopes on each site's vegetation types. In total, 10 study plots were established in

pairs in Albarella and Caleri (Figure 1). Sampling points 1 through 8 correspond to 4 ecosystems with increasing distances from the sea: beach = points 1–2 (beach without plant), dune = points 3–4 (indicator species: *Cakile maritima, Elymus farctus, Ammophila arenaria*), shrub = points 5–6 (*Juniperus communis, Phillyrea angustifolia, Tamarix gallica*), and forest = points 7–8 (*Quercus ilex, Pinus pinaster, Ulmus minor*). Sites 9 and 10 correspond to ecosystems particular to Porto Caleri only: C9 (wet lawn with *Limonium* spp. in a saltmarsh) and C10 (*Salicornia veneta*-dominated halophytic vegetation in shallow salt water) and of Albarella: A9 (new plantation, at the forest level, with protective nets around small trees of *Pinus pinea* and *Quercus ilex*) and A10 (flowerbed).

In each of these 10 points, we sampled:

- Soil microorganisms: 3 replicates a few meters away from each other, using a brass cylinder 10 cm long and 1.3 cm in diameter, at a 0–10 cm depth, discarding the litter layer when present. We collected a total of 10 (points) × 2 (sites) × 3 (replicates) = 60 replicates to be submitted to DNA extraction; these 60 replicates belong to 20 sampling points;
- Soil organic carbon: 6 cylindrical soil cores, 12 inches (=30.48 cm) long and 1 inch (=2.54 cm) in diameter, at 0–30 cm in the soil, with litter layer when present, scattered in the same areas investigated for microorganisms. We gathered the 6 soil cores (replicates) collected in each point in a single bag, obtaining a total of 10 (points) × 2 (sites) = 20 samples of soil.

In the text, for clarity we will refer to "sites" as the island of Albarella and the reserve of Caleri; "samples" as the 10 sampling points performed at each of the sites; and "ecosystems" as the points grouped two by two as in Figure 1 (colors: blue, green, yellow, orange, or gray).

*2.2. DNA Extraction, Sequencing, and Bioinformatics*

After air drying sampled soil for 3 days at room temperature, total DNA was extracted using the PowerSoil DNA isolation kit (MO BIO Laboratories Inc., CA, USA) according to the manufacturer's instructions. For the identification of fungi, the internal transcribed spacer 1 (ITS1) was amplified using the primer ITS1F forward (5′-CTTGGTCATTTAGAGGAAGTAA-3′) [65] and ITS2 reverse (5′-GCTGCGTTCTTCATCGAT GC-3′) [66]. The primer 515F forward (5′-GTGYCAGCMGCCGCGGTAA-3′) and the 806 R reverse (5′-GGACTACNVGGGTWTCTAAT-3) [67] with degenerate bases suggested by [68] and by [69] were used for the identification of the bacterial and archaeal V4 region of 16 S rDNA. DNA purification, indexing, quantification, and library preparation for the Illumina MiSeq sequencing platform (PE300) were carried out as described by Coller et al. (2019) [70]. Sequencing was carried out on an Illumina® MiSeq (PE300) platform (MiSeq Control Software 2.5.0.5 and Real-Time Analysis software 1.18.54.0; Illumina Technologies, San Diego, CA, USA).

Sequencing and bioinformatics analyses for fungi and bacteria were performed according to Coller et al. (2019) [70], while Operational Taxonomic Units (OTUs) assignment was performed using the MICCA pipeline proposed by Albanese et al. (2015) [71]; OTUs were clustered at 97% similarity cut-off, and taxonomic prediction (or OTUs assignment) was performed according to Coller et al. (2019) [70]. The assignment was based on the RDP classifier v.2.11 (https://rdp.cme.msu.edu/; accessed on 29 April 2021) and the UNITE database (https://unite.ut.ee/#main; accessed on 29 April 2021).

The efficiency of sampling structure in terms of richness was evaluated by analyzing the cumulated number of taxa detected against the number of individuals collected using accumulation models [72], and rarefaction analyses were performed according to [71].

The BioProject accession number for these SRA data is: PRJNA819224; https://www.ncbi.nlm.nih.gov/sra/PRJNA819224. accessed on 29 April 2021.

*2.3. Soil Chemical Analysis*

To estimates the organic resources available to soil living beings, we used an Australian packaged prototype. After a long calibration phase, we were able to measure data on the organic carbon content in the first 30 cm of soil comparable to others found in the laboratory

with another loss on ignition method. It was very important to align the sampling points along a gradient from the sea to the land to identify ecosystems of different ages in terms of evolution. In each system we collected 6 soil cores using an AMS EZ Eject Soil Probes cylindrical sampler (soil sample size: diameter: 1″ = 2.54 cm; length: 12″ = 30.48 cm). These soil cores were harvested in an area of 100 square meters in order to correspond, on average, to the first 30 cm of the soil (soil classification: Arenosols, IUSS Working Group WRB et al., (2015) [73]) of each ecosystem. Soil litter was cut with the circular tip of the tool and ended up occupying a space at the top and within 30 cm of the carrot. Taken to the laboratory, the mixed samples were air-dried for 2 weeks. A 2 mm net sieve made it possible to extract pieces of organic matter (litter) and pebbles (skeleton) from the soil. Soil ≤ 2 mm, litter, and skeleton were weighed separately.

The soil organic carbon content in soil ≤ 2 mm was measured by two ignition methods, using Skalar Primacs ATC-100-IC-E [74] and Carbon Still [75].

Skalar Primacs uses a measuring principle in accordance with DIN 19539: 1 g of soil is gradually brought from 150 to 400 °C in 480 s, with a controlled temperature increase of 70 °C per minute. An IR detection system of the emitted $CO_2$ flow returns the measurement of the carbon lost in percentage content with respect to the weight of the sample, and then converts it into grams considering, in our specific case, the bulk density of a sandy soil (1.6 g cm$^{-3}$) as representative of this coastal environment.

Carbon Still has been designed for up to 2000 g soil samples. We used samples of 600–650 g. Soil is brought to 120 °C and left for 10 min at this temperature to dehydrate before being weighed for a first time. Hot air at a controlled temperature is injected into the sample until it reaches the self-combustion temperature of the carbon (about 350 °C). At this point, temperature rises by itself in the sample up to around 400 °C. Once all the organic carbon oxidizes, the temperature drops, and it is possible to reweigh the sample. The estimate of the carbon lost from the sample is made by the difference in the weight before and after the controlled burning. The more organic matter in the soil, the higher the temperature peak reached in the chamber (400 ± 25 °C). Using a standard peak temperature for all samples would distort the measurements. Soil carbon data were expressed per hectare, knowing that the 6 carrots had an area of (6 • pi • (2.54/2)ˆ2 = 30.40245 cm$^2$).

To estimate the organic carbon content in the litter. we removed the weight of the water (15%) left in the air-dried litter, and then divided it by 2 (organic carbon weight = half that of dry organic matter).

In the 10 Albarella + 10 Caleri soil samples, the microplastic content was also investigated. The survey methodology was as follows [76]: ca. 2 g sample + 10 mL 15% $H_2O_2$; 5 min ultrasonic bath; +40 μL 1 M HCl; 60 min 90 °C water bath; cool; 2 mL suspension + 3.4 g $ZnCl_2$; dissolve; spin at ca. 1000 g for 15 min; 1 mL supernatant + 1 mL water for microscopy 20 μL dilution; covered with glass, bright field (and rarely phase contrast); screening with 100× mag, count with 400× mag.

### 2.4. Statistical Analysis

Taxonomic alpha diversity was calculated as estimated community diversity by the Shannon index and Chao 1 index. The alpha diversity indices were analyzed by one-way analysis of variance (ANOVA, once normal distribution and homoscedasticity were controlled) to determine significant differences among the samples. Carried out using the 'pcoa' function in R package "ape", a principal coordinates analysis (PcoA) was selected to illustrate the clustering of different samples. We also used the permutational multivariate analysis of variance (PERMANOVA) to determine the distances of each sample to the group centroid in a PcoA and to provide a *p*-value for the significance of the grouping. PERMANOVA was implemented using the 'adonis' function in the R 'vegan' package.

We used the null-model-based approaches, which have been widely used in microbial ecology studies [77–79], to estimate the influences of ecological processes on the community assembly of soil bacteria and fungi. We tested for the phylogenetic signal to determine whether we could use phylogenetic turnover to make ecological inferences in our system by

using the R 'microeco' package. Then, following the analytical procedure developed by [78], we first calculated the pairwise phylogenetic turnover (βMNTD) in the two islands by using the 'comdistnt' function in the R 'picante' package. Then, randomization was used to generate a null distribution of phylogenetic turnover. The value of βNTI characterizes the magnitude of deviation between observed βMNTD and the mean of the null distribution of βMNTD. As part of the second major step in the procedure, we calculated standardized taxonomic beta diversity—the Bray–Curtis-based Raup–Crick metric ($RC_{Bray}$) based on the difference between the observed Bray–Curtis dissimilarity and its null distribution. Combining the value of βNTI and $RC_{Bray}$, we could infer the potential mechanisms of community assembly: βNTI that fell below −2 and above 2 indicates the homogeneous selection and variable selection, respectively; $|βNTI| < 2$ and $RC_{bray} > 0.95$ indicate dispersal limitation; $|βNTI| < 2$ and $RC_{Bray} < −0.95$ indicate homogenizing dispersal; $|βNTI| < 2$ and $|RC_{Bray}| < 0.95$ indicate that community turnovers are assessed as the effect of "undominated" assembly, which mainly contains weak dispersal, weak selection, diversification, and/or drift. The relative contribution of the deterministic process was calculated as the sum of homogeneous selection and variable selection; the stochastic process was calculated as the sum of homogenizing dispersal and dispersal limitation [78,80]. Community assembly processes were evaluated by applying the null model in the "picante" package in R software.

### 3. Results: Are the Soil Microorganisms Different along the Vegetation Series and between Anthropized and Natural Sites?

*3.1. Soil Bacteria and Fungi*

From a total of 33,168,400 filter-passing reads within the project, 52,756 bacterial OTUs and 11,676 fungal OTUs were obtained across all 60 samples examined. In the following description, samples from the anthropized place (Albarella) have a code starting with A, while the natural reserve of Caleri is coded with C. Numbers 1 through 8 of each sample increase along the transect from shore to forest. Samples 9 and 10 for both places are unique, nonmatching biotopes.

The bacterial community composition at the phylum level (relative abundance > 1%) is shown in Figure 2A. The dominant phyla in all soil samples were Proteobacteria, Acidobacteria, and Planctomycetes. The relative abundance of bacteria in the same ecosystem (1 ecosystem = 2 phytocoenosis = 2 sampling points = $2 \times 3 = 6$ repetitions = 6 samples) did not result very differently. Other than some unidentified phyla, the fungal OTUs collected from all 60 samples were primarily classified into six phyla (Figure 2B). Ascomycota was the most dominant phylum across all samples except for A7 and C10, accounting for more than 50% of the sequences in the samples collected from the beach and dune area, and accounting for 25–70% of the sequences in the shrub, forest, forest plantation, flowerbed, and wetland. It can be noted that Basidiomycota dominated the A7 sample accounting for 63.44%, and also displayed high abundance in the A5, C5, C6, and A8 samples accounting for 30.34–39.53%. Furthermore, the Zygomycota phylum was rare in the beach and wetland and had the highest abundance (29.37%) in A6 of the Albarella shrub sampling point.

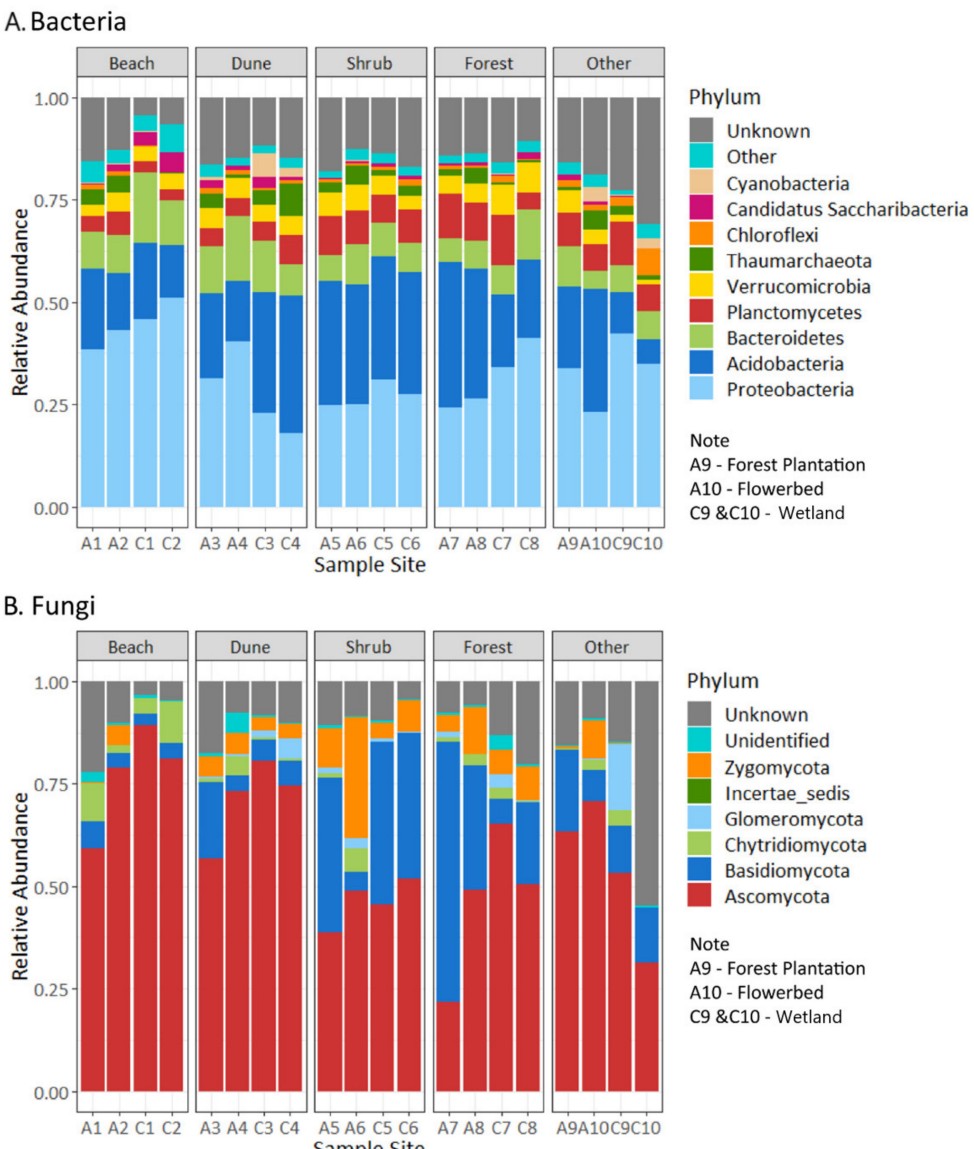

**Figure 2.** Phylum-level abundance from Albarella and Caleri. The taxa represented accounted for >1% abundance in at least one sampling point. Each ecosystem (beach, dune, shrub, forest, and other) is represented by two phytocoenosises corresponding to two sampling points and (with three repetitions in each point): $2 \times 3 = 6$ samples.

The relative abundance, which is visualized in Figure 2, shows the composition of microorganisms in the soil samples. Adding an error bar in the graph was less telling than a table (Table 1). Table 1 shows the probability that the taxa at the phylum level can belong to a single population by comparing the sites (Albarella to Caleri) and the samples (10 samples at Albarella to 10 samples at Caleri). If the samples are compared, it is clearly seen that many of these taxa are not specific to Albarella or Caleri but belong to distinct populations (samples in the same ecosystems: ecosystem 1 in Albarella against ecosystem 1 in caleri, ecosystem 2 in Albarella against ecosystem 2 in Caleri, up to ecosystem 10 Albarella against ecosystem 10 Caleri). This means that human action on ecosystems may have disrupted combinations of the same microorganisms by aggregating them differently into apparently similar ecosystems.

**Table 1.** Probability (*p*-value) that taxa at phylum level can belong to the same population comparing Sites (Albarella and Caleri) or Samples (10 samples in each site, taken 2 by 2).

| Variable | Source | Sig. (*p*-Value) |
|---|---|---|
| Bacteria | | |
| Proteobacteria | Site | 0.654 |
| | Sample | 0.001 |
| Acidobacteria | Site | 0.116 |
| | Sample | 0.000 |
| Bacteroidetes | Site | 0.687 |
| | Sample | 0.004 |
| Planctomycetes | Site | 0.286 |
| | Sample | 0.000 |
| Verrucomicrobia | Site | 0.034 |
| | Sample | 0.000 |
| Thaumarchaeota | Site | 0.020 |
| | Sample | 0.001 |
| Chloroflexi | Site | 0.314 |
| | Sample | 0.000 |
| Candidatus Saccharibacteria | Site | 0.432 |
| | Sample | 0.000 |
| Cyanobacteria/Chloroplast | Site | 0.449 |
| | Sample | 0.001 |
| Fungi | | |
| Ascomycota | Site | 0.602 |
| | Sample | 0.001 |
| Basidiomycota | Site | 0.633 |
| | Sample | 0.002 |
| Chytridiomycota | Site | 0.007 |
| | Sample | 0.011 |
| Glomeromycota | Site | 0.375 |
| | Sample | 0.000 |
| Zygomycota | Site | 0.003 |
| | Sample | 0.000 |

Principal coordinates analysis (PcoA) revealed the similarity of bacteria and fungi composition in the soils from Albarella and Caleri. Soil samples from similar ecosystems (beach, dune, shrub, and forest) showed similar compositions (Figure 3). It should be noted that soil bacterial and fungal communities in the wetland habitat (Caleri C9 and C10) were distant from other communities. The separation of these samples in wet, salty soils from the others takes up all the variance of axis 1. Then, the different samples are arranged from the top to the bottom of axis 2, passing from the poorly evolved stages on the sandy beach, toward the intermediate herbaceous and shrub dunes, and finishing in the inland forest. The samples from Caleri (natural, blue) and Albarella (anthropized, red) run side by side in the same direction. The planting of new trees (A9) and the flower beds (A10) of Albarella end up in the middle of the series of points.

With the fungal component, the gradient from the sea (beach, less developed stage) toward the mainland (forest, more advanced stage) is evident both for Caleri and Albarella, from right to left on axis 1; axis 2 contrasts the samples in the wet and salty, and, particularly, Caleri soils from all the others. It is interesting to note that the variance explained of the two axes of the PcoA by the genetic difference of the bacteria is higher than that of the fungi. It is also interesting to note that the axes are inverted: for bacteria, the first separates OTUs in water (with a change of energy metabolism from aerobiosis to anaerobiosis) from the others, and the second marks the separation from the sea to the mainland; the opposite happens for fungi, which confirm to be organisms less sensitive to osmolarity stress from brackish waters and more reactive to the change of phytocoenosis. The assemblage compositions of soil bacteria and fungi differed significantly across the sample sites, Albarella or Caleri (PERMANOVA: $R^2 = 0.777$ for bacteria, $R^2 = 0.709$ for fungi, $p < 0.001$ in both cases).

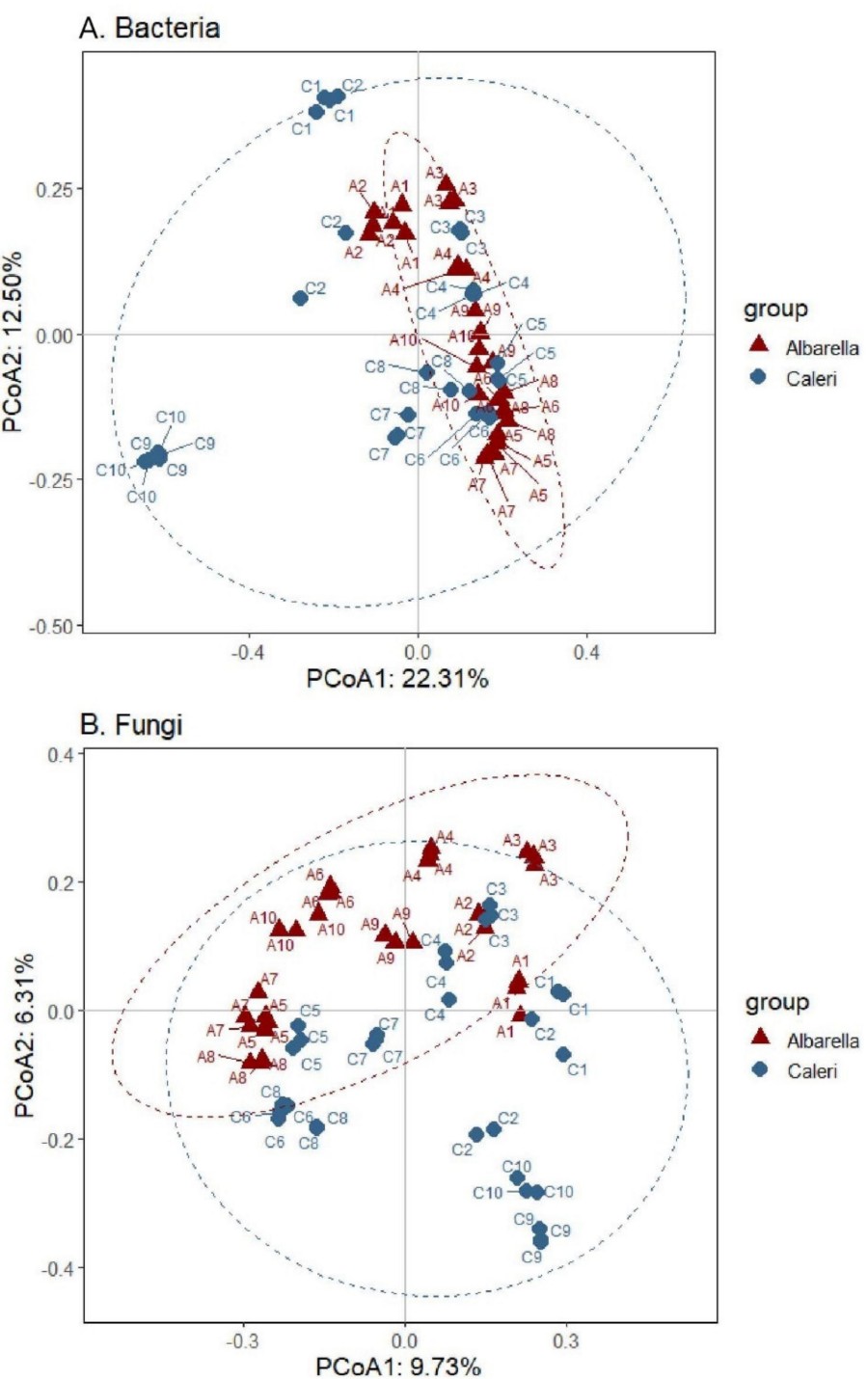

**Figure 3.** PcoA of microbial community composition across all soil samples in Albarella and Caleri.

*3.2. Soil Bacteria and Fungi Shannon's and Chao1's Diversities*

    The OTU level classification (97% DNA collinearity, which is acknowledged to be near-species level) was used to calculate the Shannon–Wiener (*H'*) and Chao1 main indices. We focused on the "essence" of biodiversity rather than on the evenness index (which is more a derivative of *H'*). In samples with the same number of species:

- The Shannon–Wiener index is the highest when individuals have the same frequency in each "species = sampled community" (high evenness); $H'$ is the lowest when all "species" but one is represented by a single individual, and one "species" cumulated all different individuals (low evenness).
- The Chao1 index is high if the species represented by a single individual dominate in comparison to those represented by two individuals; the number of individuals (one or two) representing the species plays a great role (low evenness). The Chao1 index is low when the number of species represented by a single individual is similar to that of species represented by two individuals (high evenness).

With respect to the bacterial communities, both the Shannon–Wiener and Chao1 indices significantly differ between Albarella and Caleri (ANOVA: respectively, $p < 0.002$, $p < 0.001$, Table 1). For those represented by fungal communities with a lower number of OTUs, the Shannon–Wiener index value was shared by Albarella and Caleri sites ($p = 0.003$), but this was not the case with Chao1 ($p = 0.834$, Table 2).

**Table 2.** Difference significance for whole sites or sample level on microbial diversity and richness (ANOVA).

| Variable | Source | df | Mean Square | F-Value | Sig. ($p$-Value) |
|---|---|---|---|---|---|
| Bacteria | | | | | |
| Shannon–Wiener | Site | 1 | 1.7576 | 9.944 | 0.002 |
| | Sample | 19 | 0.5776 | 22.35 | 0.000 |
| Chao1 | Site | 1 | 11,121,957 | 11.95 | 0.001 |
| | Sample | 19 | 2,935,996 | 12.61 | 0.000 |
| Fungi | | | | | |
| Shannon–Wiener | Site | 1 | 3.970 | 9.32 | 0.003 |
| | Sample | 19 | 1.2612 | 10.7 | 0.000 |
| Chao1 | Site | 1 | 1780 | 0.044 | 0.834 |
| | Sample | 19 | 104,903 | 12 | 0.000 |

Differences were also observed between the soil samples ($p < 0.000$, Table 2). For bacterial and fungal communities, the Shannon–Wiener and Chao1 indices grew with the increasing complexity of ecosystems, from the soil with pioneer vegetation on the beach, toward the herbaceous plant formations on the white dune, to the shrub areas on the gray dune, and ending in the multilevel forest on paleo-dunes (Figures 4 and 5). Overall, for bacterial communities' soils from Caleri, C1 and C2 (beach without or with low pioneer vegetation) had the lowest Shannon–Wiener (5.5) and Chao1 indices (1500). Instead, the highest of them were observed in Albarella, in A9 (new plantation of trees), where the values reached 7.2 and 5200, and in A10 (flowerbed), where they arrived at 7 and 7400, respectively (Figure 4A,B).

For the fungal populations, albeit with lower values, we observed the same trend of the indices dependent on the complexity of the ecosystems, increasing from the beach pioneer cover to the internal multistate forest (Shannon–Wiener from 2.5 to 3.5 and Chao1 from 200 to 500).

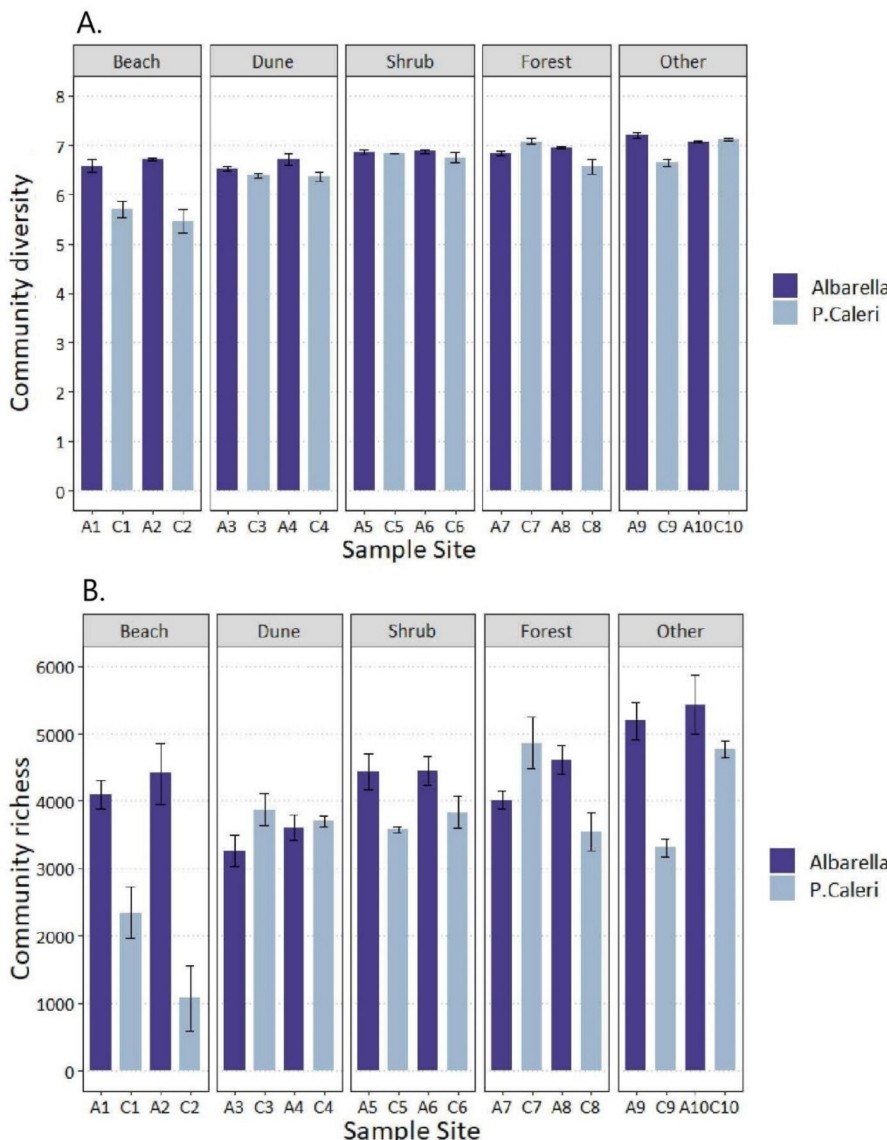

**Figure 4.** Soil bacteria diversity at the I level. Values shown in the figure are means of (**A**) Shannon–Wiener and (**B**) Chao1 indices; vertical bars are SE (n = 3).

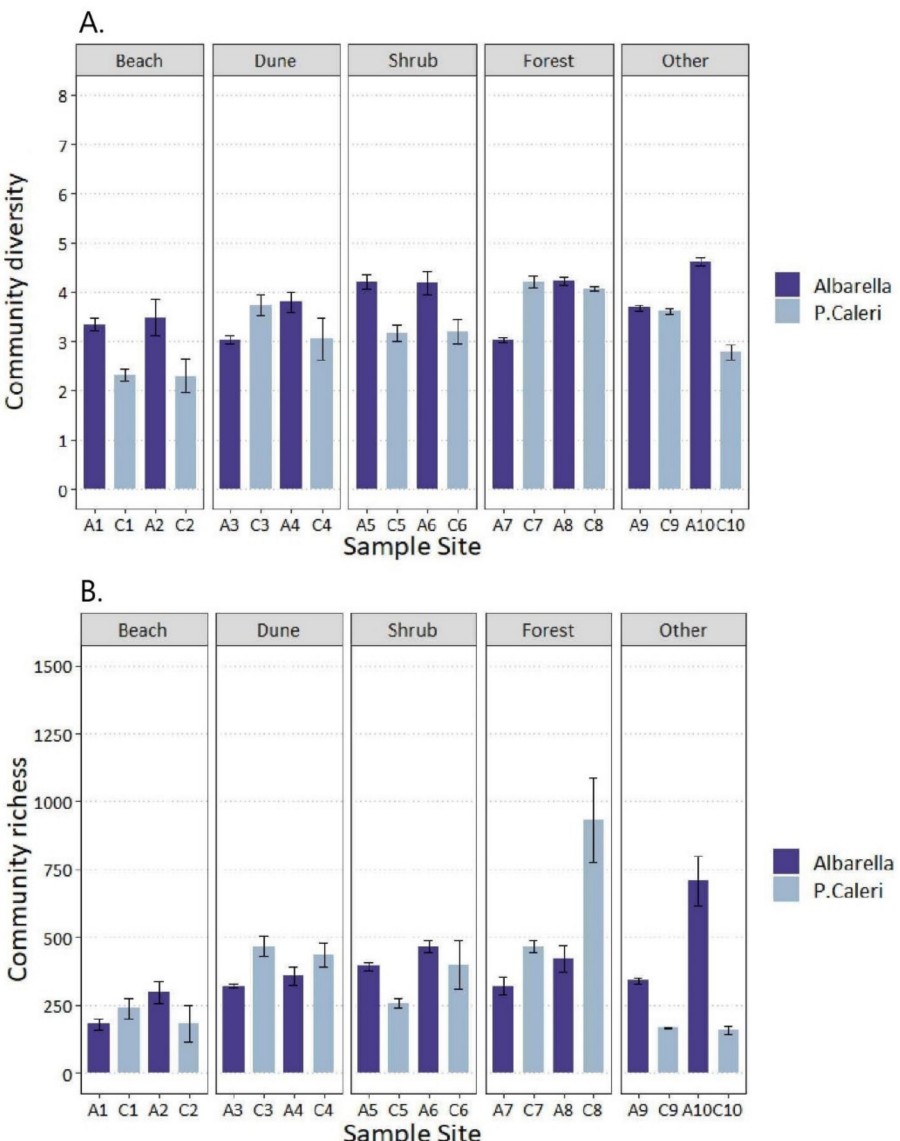

**Figure 5.** Soil fungal diversity at tIOTU level. Values shown in the figure are means of (**A**) Shannon–Wiener and (**B**) Chao1; vertical bars are SE (n = 3).

### 3.3. Ecological Processes Governing Bacterial and Fungal Community Assembly

We found a significant phylogenetic signal for bacteria and fungi in Albarella and Caleri, but a phylogenetic signal for bacteria only across relatively short phylogenetic distances (Figure 6). In order to understand how communities of microorganisms differ from each other, we hypothesized that:

(1) Naturally assembled microorganisms living in nearby sampling points could have more affinity with each other and be better coordinated by necessity in the exploitation of resources;

(2) That these natural assemblages of microorganisms were better adapted to live together than microorganisms grouped by chance following human intervention;

(3) That the affinity of natural groupings had a strong probability of being the expression of an underlying phylogenesis.

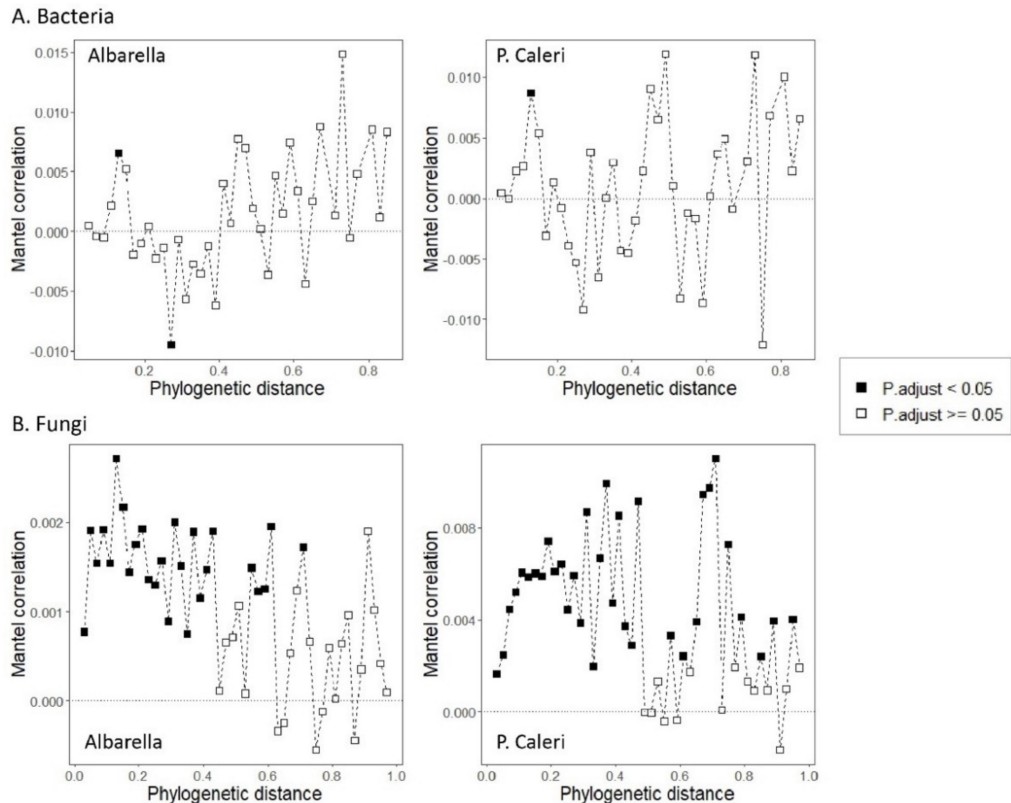

**Figure 6.** Phylogenetic Mantel correlogram showing significant phylogenetic signal across phylogenetic distances. Solid and open symbols denote significant and nonsignificant correlations, respectively, relating betwen-OTU niche differences to between-OTU phylogenetic distances, across a given phylogenetic distance.

The relative contributions of ecological processes varied in the bacteria and fungi (Figures 7 and 8).

In Albarella, 12.46% of the samples were less than the expected phylogenetic turnover ($\beta$NTI < −2, Figure 7a), while 28.32% of the phylogenetic turnover of soil bacteria were significantly more than expected phylogenetic turnover ($\beta$NTI > 2, Figure 7a), and 59.22% of the cases were not significantly different from the null expectation (|$\beta$NTI| < 2, Figure 7a).

Most of the phylogenetic turnover of soil bacteria across samples of Caleri was significantly different from the null expectation ($\beta$NTI < −2 in 17.01% cases, $\beta$NTI > 2 in 42.20% cases, |$\beta$NTI| < 2 in 40.79% cases, Figure 7c). In the cases that were not significantly different from the null expectation, most exhibited a larger taxonomic turnover than expected by chance (RCBray > 0.95, Figure 7b,d). Overall, these results suggest that selection, particularly variable selection, may have dominated the community assembly of soil bacteria in Caleri, while homogenizing dispersal may have played an important role in causing taxonomic turnover in Albarella (Figure 7e).

As for soil fungi, 40.92% and 27.39% of the sample were less than the expected phylogenetic turnover in Albarella and Caleri, respectively ($\beta$NTI < −2, Figure 8a,c). Phylogenetic turnover between most samples within Albarella (58.62%) and Caleri (68.52%) did not differ from the null expectation (|$\beta$NTI| < 2, Figure 8a,c), and among these samples, most of their taxonomic turnover was significantly greater than expected by chance (RCBray > 0.95, Figure 8b,d), suggesting dispersal limitation to be the main driver for fungal taxonomic turnover within these two islands (Figure 8e).

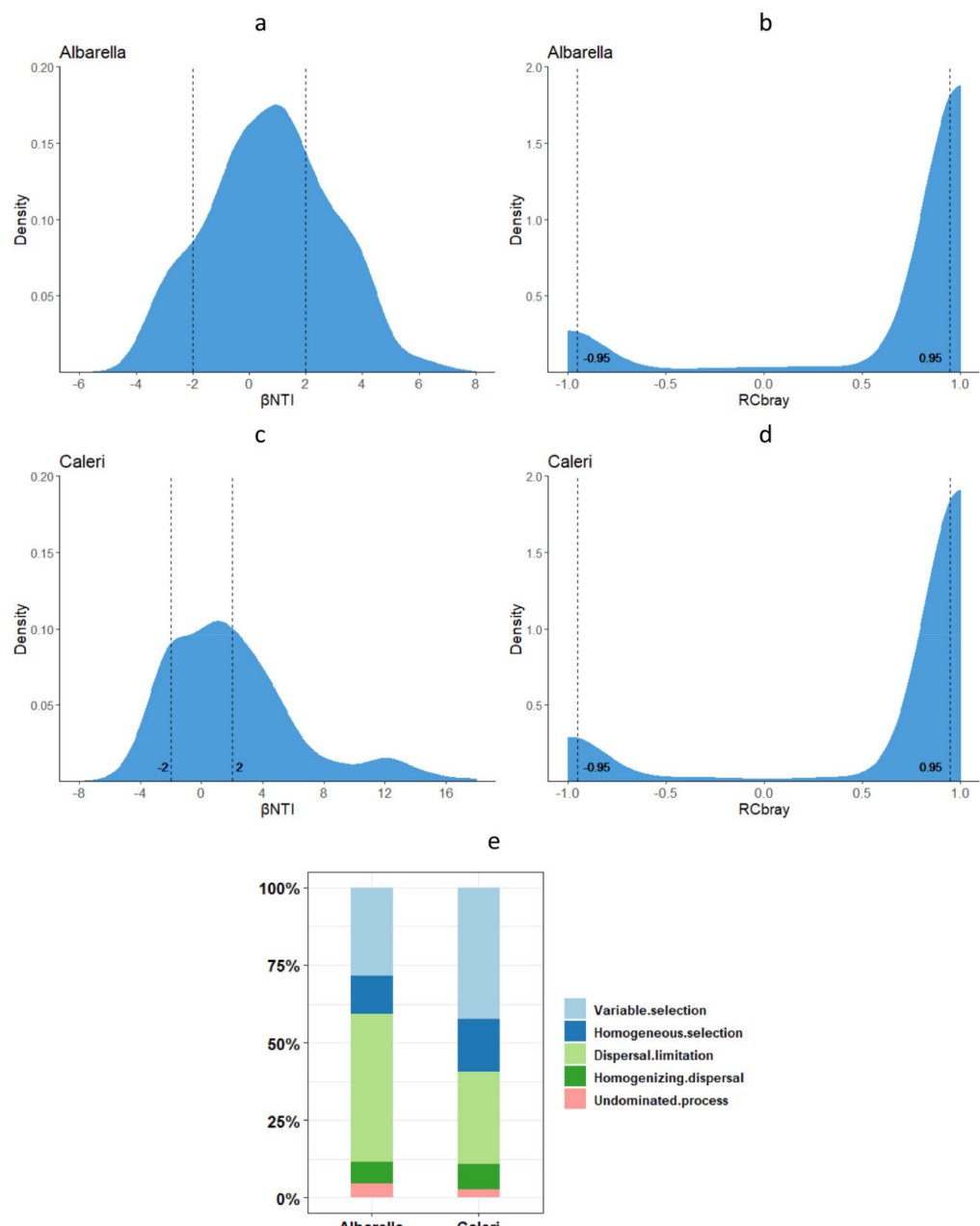

**Figure 7.** Distribution of standardized phylogenetic turnover and taxonomic turnover of soil bacteria within Albarella and Caleri. (**a**): βNTI of Albarella, (**b**): RCBray of Albarella, (**c**): βNTI of Caleri, (**d**): RCBray of Caleri, (**e**): percentages of the five structuring processes in Albarella and Caleri.

We found a higher significant phylogenetic signal in Caleri (more natural) than in Albarella (more anthropized). Additionally, that variable selection may have dominated the community assembly of soil bacteria in Caleri, while homogenizing dispersal may have played an important role in causing taxonomic turnover in Albarella. In the last 100 years, the actions on the vegetation cover and the changes in the landscape structure were more impactful in Albarella than in Caleri. We know that for bacteria, the times of vertical chromosomal evolution are much longer [81]. How could these changes also have affected the detected phylogenetic signal?

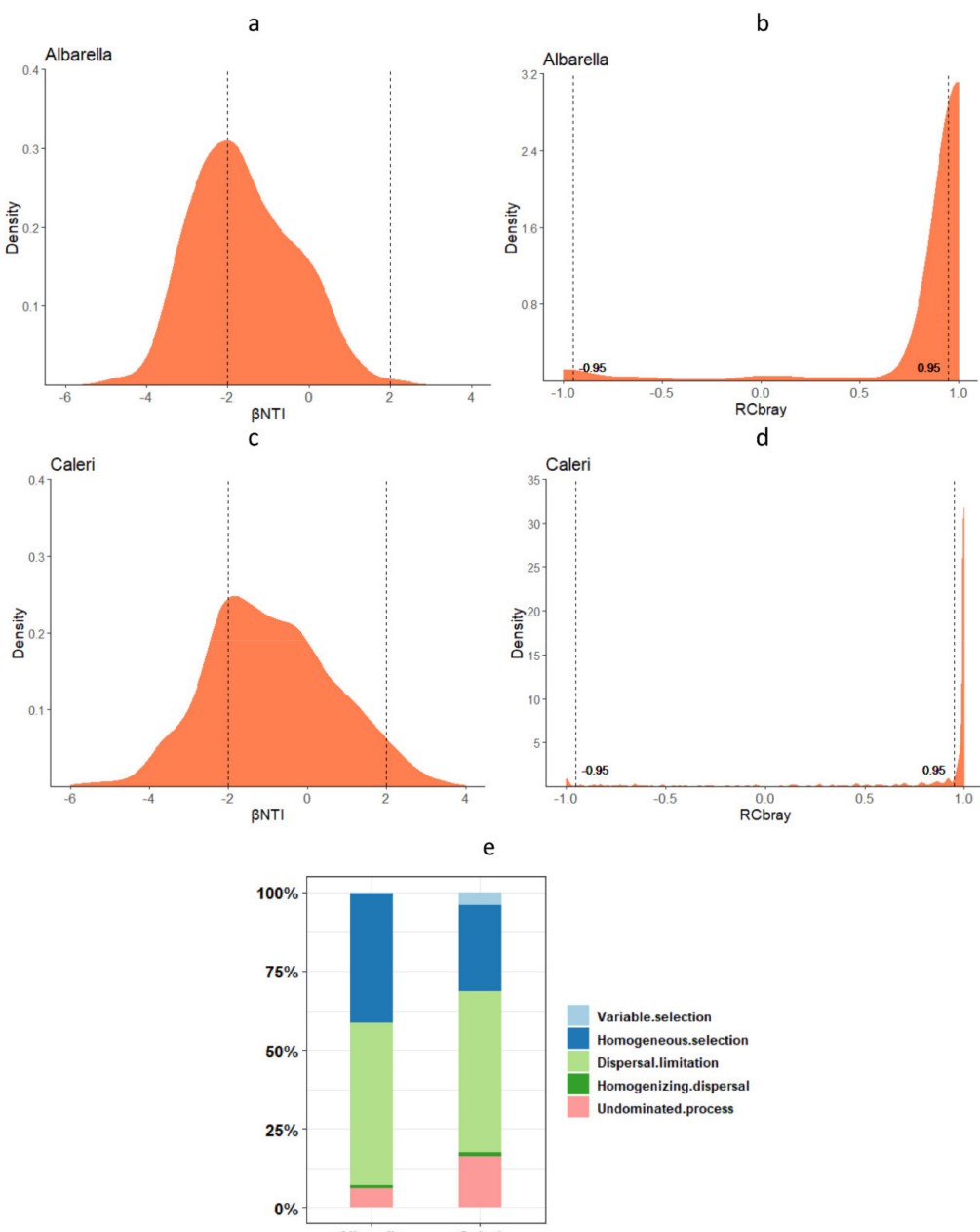

**Figure 8.** Distribution of standardized phylogenetic turnover and taxonomic turnover of soil fungi within Albarella and Caleri. (**a**): βNTI of Albarella, (**b**): RCBray of Albarella, (**c**): βNTI of Caleri, (**d**): RCBray of Caleri, (**e**): percentages of the five structuring processes in Albarella and Caleri.

Horizontal gene transfer and illegitimate recombination events could have allowed the bacteria which reproduce in a more natural environment to respond in a more collaborative way to the selection of the environment, compared to bacteria reproducing in a less natural setting. This fact could have produced a higher significant phylogeny; from a purely chemical–physical point of view of cellular contents, a genetic kinship is probably indispensable for collaboration in the exploitation of resources. This line of thought leads to the notions so well presented and proven by Lynn Margulis in her book [1], and which lend credence to the still controversial, but fascinating, Gaia hypothesis [82].

### 3.4. Plastics

We understand that this is a rough estimate. With the measurements made (Figure 9), it seems reasonable to say that the more man-made soils of Albarella have a microplastic content no different from the more natural ones of Caleri. We can also say that the values found are similar to the minimum values in European agricultural soils [76].

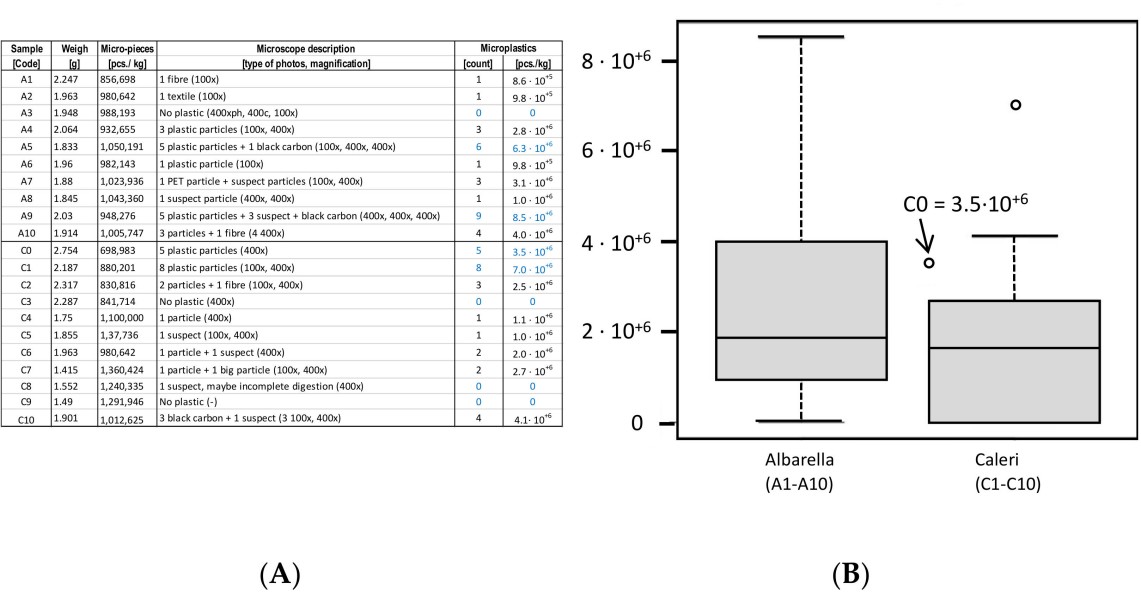

| Sample [Code] | Weigh [g] | Micro-pieces [pcs./ kg] | Microscope description [type of photos, magnification] | Microplastics [count] | [pcs./kg] |
|---|---|---|---|---|---|
| A1 | 2.247 | 856,698 | 1 fibre (100x) | 1 | 8.6 · 10^{+5} |
| A2 | 1.963 | 980,642 | 1 textile (100x) | 1 | 9.8 · 10^{+5} |
| A3 | 1.948 | 988,193 | No plastic (400xph, 400c, 100x) | 0 | 0 |
| A4 | 2.064 | 932,655 | 3 plastic particles (100x, 400x) | 3 | 2.8 · 10^{+6} |
| A5 | 1.833 | 1,050,191 | 5 plastic particles + 1 black carbon (100x, 400x, 400x) | 6 | 6.3 · 10^{+6} |
| A6 | 1.96 | 982,143 | 1 plastic particle (100x) | 1 | 9.8 · 10^{+5} |
| A7 | 1.88 | 1,023,936 | 1 PET particle + suspect particles (100x, 400x) | 3 | 3.1 · 10^{+6} |
| A8 | 1.845 | 1,043,360 | 1 suspect particle (400x, 400x) | 1 | 1.0 · 10^{+6} |
| A9 | 2.03 | 948,276 | 5 plastic particles + 3 suspect + black carbon (400x, 400x, 400x) | 9 | 8.5 · 10^{+6} |
| A10 | 1.914 | 1,005,747 | 3 particles + 1 fibre (4 400x) | 4 | 4.0 · 10^{+6} |
| C0 | 2.754 | 698,983 | 5 plastic particles (400x) | 5 | 3.5 · 10^{+6} |
| C1 | 2.187 | 880,201 | 8 plastic particles (100x, 400x) | 8 | 7.0 · 10^{+6} |
| C2 | 2.317 | 830,816 | 2 particles + 1 fibre (100x, 400x) | 3 | 2.5 · 10^{+6} |
| C3 | 2.287 | 841,714 | No plastic (400x) | 0 | 0 |
| C4 | 1.75 | 1,100,000 | 1 particle (400x) | 1 | 1.1 · 10^{+6} |
| C5 | 1.855 | 1,37,736 | 1 suspect (100x, 400x) | 1 | 1.0 · 10^{+6} |
| C6 | 1.963 | 980,642 | 1 particle + 1 suspect (400x) | 2 | 2.0 · 10^{+6} |
| C7 | 1.415 | 1,360,424 | 1 particle + 1 big particle (100x, 400x) | 2 | 2.7 · 10^{+6} |
| C8 | 1.552 | 1,240,335 | 1 suspect, maybe incomplete digestion (400x) | 0 | 0 |
| C9 | 1.49 | 1,291,946 | No plastic (-) | 0 | 0 |
| C10 | 1.901 | 1,012,625 | 3 black carbon + 1 suspect (3 100x, 400x) | 4 | 4.1 · 10^{+6} |

F value: 0.53; *p* (<F): 0.46

C0 = 3.5·10^{+6}

Albarella (A1-A10)     Caleri (C1-C10)

(**A**)                (**B**)

**Figure 9.** Microplastic content (pieces per kg; pcs./kg) in the soil samples of Albarella and Caleri. Table on the left (**A**) and boxplot on the right (**B**): median, first quartile in the boxes, second quartile within the whiskers, and extreme values. Numbers in Blue are comparably safe, single count can be questioned, all low numbers are not very accurate. C0 = first 5 cm of soil under 50 cm of water, low tide; collection made in Caleri, at a point 50 m from Albarella.

### 3.5. Carbon Still or Skalar Primacs Soil Organic Carbon Measurements?

We found a good regression line between the carbon estimated with the Carbon Still and that found with the Skalar Primacs (Figure 10). Designed to help Australian farmers to restore their soils [75], the Carbon Still returns reliable results and may help mitigate climate warming.

Even if the estimation process considers two samples of very different masses, 650 g of soil with the Carbon Still and 1 g with the Primacs ATC, we found a good linear relationship between the values estimated with the two methods (Figure 10A). Note how the addition of the estimated organic carbon in the litter (remains that did not pass the 2 mm sieve) to the organic carbon value measured with the Carbon Still barely improved the variance explained by the straight line (R2 0.850 -> 0.857). The slope of the line instead changes a lot (1.0923 -> 1.739) and tells us that the litter can also correspond to 40% of the soil carbon. The humus forms which characterize the soil in Albarella and Caleri vary from Mull to Moder [83–85]. Even if in Mull forms the litter thickness is low and changes along the year, we can consider that at least half of the litter organic carbon (20–25 t/ha) is always present in Albarella and Caleri soils. The localized variability of the litter thickness and the consequent measurements obtained with the Carbon Still can also be seen on the graph, since these are less close to the regression line than those obtained with the Sakalar Primacs method. For this reason, we considered Skalar Primacs TOC400 in the following analyzes, although estimating the localized contribution of different thicknesses of litter would be a more correct practice, even if this requires a greater number of samples.

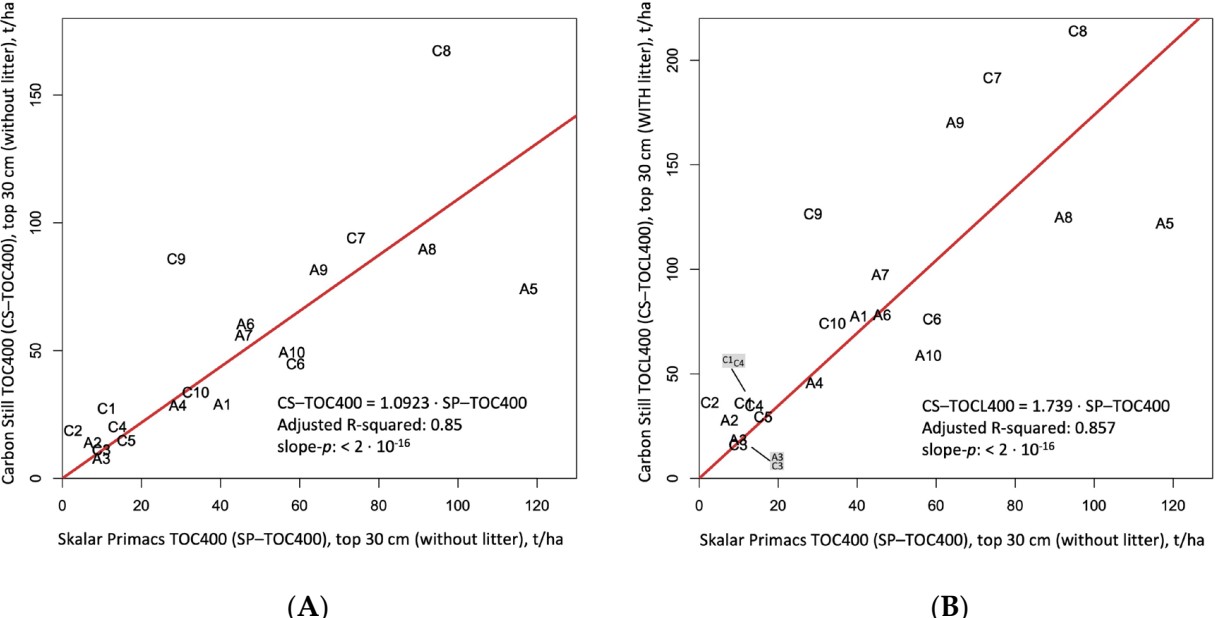

**Figure 10.** (**A**) Linear regressions between organic carbon estimated in organic mineral soil samples with Carbon Still (CS–TOC400) and Skalar Primacs (SP–TOC400). (**B**) Linear regressions between organic carbon estimated in organic mineral soil plus in litter samples with Carbon Still (CS–TOCL400) and organic carbon estimated in organic mineral soil samples with Skalar Primacs (SP–TOC400). Sliding from left to right on the graphs you go from the beach into the forest at the two sites (A = Albarella; C = Caleri): 1–2 = beach; 3–4 = dune; 5–6 = shrubland; 7–8 = forest; 9–10 = other (two humid areas in Caleri; flower bed, and new plantation in Albarella).

### 3.6. Biodiversity and Soil Organic Carbon Gradient 1

Sliding from left to right on the graphs (Figure 11), we go from the beach into the forest in both Albarella and Caleri sites. Notice how SP-TOC400 increases progressively in the same direction. The lowest SOC contents are in ecosystems close to the sea, beach, or embryonic dune (C2, C3, A2, and A3). Young plantations in Albarella (A9) and wetland in Caleri (C9) show a medium-high content of SOC. The highest SOC levels were measured in Mediterranean scrub or woodland ecosystems (C7, C8, A5, and A8). The biodiversity recorded along the range from sea to land also grows, as estimated by the Shannon and Chao1 indices.

These graphs clearly illustrate what we have partly presented in the previous figures, namely, that the biodiversity indices of bacteria are higher than those of fungi. The diversity of bacteria tends to be universally higher than that of fungi. Bacteria (asexual) are defined on 16 S, which has a merely mutational evolutionary way of changing, which differs from the internal transcribed spacer that we use for fungi (eukaryotes which can rely also on sexual recombination). Diversity indices seem to grow along the series that goes from the sea to the land and which corresponds to an increase in the complexity of ecosystems. However, even if the regression lines (separated for bacteria and fungi) between the values of the biodiversity indices and those of TOC400 have statistically good parameters, they explain only 1/3 of the distribution of the points ($0.23 < R2 < 0.42$). In the discussion, we will take up this important aspect by recalculating the regression. Note in Figure 11B, in the lower left part between the two straight lines, the bacteria C1 and C2 codes. They correspond to very low values of the Chao1 index compared to those predicted by the green rectilinear model. In the discussion part, we tried to force the model into the origin.

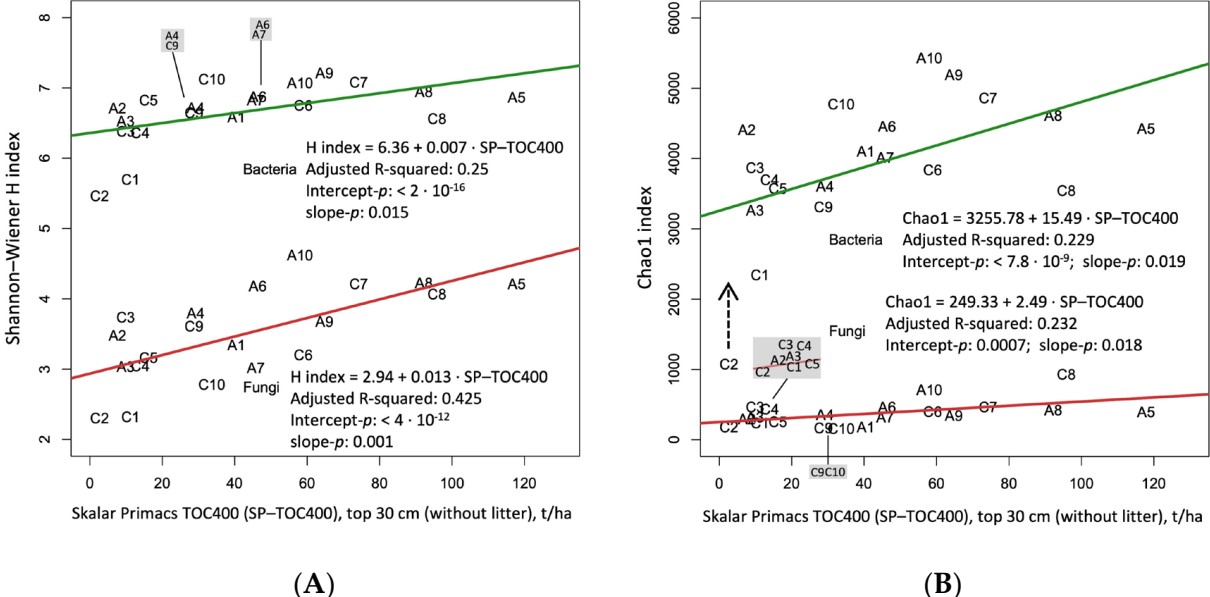

**Figure 11.** Linear regressions between (**A**) Shannon–Wiener or (**B**) Chao1 indices (OTUs of bacteria and fungi) and the Skalar Primacs TOC400 (SP–TOC400) in Albarella and Caleri. Let us remember that there is a strong linear model that links the measurements made with the Skalar Primacs (SP–TOC400) with those made with the Carbon Still (CS–TOC400) (Figure 10A) and (CS–TOCL400), also with the carbon contained in the litter, (Figure 10B).

## 4. Discussion

Let us return to the definition of biodiversity: "Biological diversity means the variability among living organisms from the ecological complexes of which organisms are part, and it is defined as species richness and relative species abundance in space and time". The first objective of this study was to investigate the difference in microbial community diversity and community structure between anthropic and natural ecosystems. Soil bacterial communities were dominated by Proteobacteria, Actinobacteria, and Bacteroidetes (Figure 2A), and soil fungal communities were dominated by Ascomycota and Basidiomycota (Figure 2B).

Management can change the structure of the bacterial communities. Comparing those of seminatural relictual forests (C7 and C8) to the ones of managed (A7 and A8) forests or even more artificial ecosystems (A9 and A10), we see that the seminatural forests are poorer in Acidobacteria (Figure 2A) and richer in Basidiomycota (Figure 2B). The vegetation of forests, besides exudation from roots, produces abundant litter and wood debris. In comparison to herbaceous vegetation, trees interact with soil more by the delivery of material destined to gradual decay and slow turnover, which can strongly influence soil properties [86]. Thus, the dominant plants of the forest exert a strong influence on the composition of soil microbial communities [87].

In this study, the bacterial and fungal diversity from different soil samples were significantly different (Table 1) when observing the lower soil bacterial diversity and richness indices in the natural beach, and the highest diversity and richness presented in new plantation (A9) and flowerbed (A10) of Albarella (Figure 4). As for the fungal community, higher diversity and richness were observed in natural forests (C8) and flowerbeds (A10). In previous studies, plantation land had higher intrinsic growth rates and heavier disturbances to soil microorganisms [88,89]. Soil fungal communities could more rapidly respond to the changes [90] and to plantation land, because they are more resilient in the disturbances; thus, fungal diversity increases in these situations [91,92]. Soil bacterial community could increase in undisturbed conditions [43,93,94]. As we observed that both the soil bacterial and fungal diversity was higher in artificial vegetation (Albarella), we carried out final

analyses on the main energy source that the soil contains, which is its organic carbon [95], although we know that this potential resource is linked to an enzymatic response that has yet to be fully unveiled [96]. We are measuring organic carbon from the sampled core, which accounts mostly for un-decomposed polymers (which would be higher under woody vegetation or submerged soil, but for reasons of slow decomposition due to chemical and physical constraints and not to active physiology of plant–microbe dynamics), and we are overlooking and missing all the low-molecular-weight root exudates that could be directly fueling microbes which could be located either in the rhizoplane–rhizosphere soil attached to roots (we are not sampling root systems) or even translocated by mycorrhizal network to deeper soil layers. Let us see what happens, then we will decide how to draw conclusions, if there are any.

### 4.1. Biodiversity and Soil Organic Carbon Gradient 2

Adopting a logarithmic scale for the values of the independent variable (SP-TOC400), the model of Figure 11 was improved. Two other tricks have also been introduced: increasing the number of points by considering bacteria and fungi together; and forcing the regression line into the origin (Figure 12).

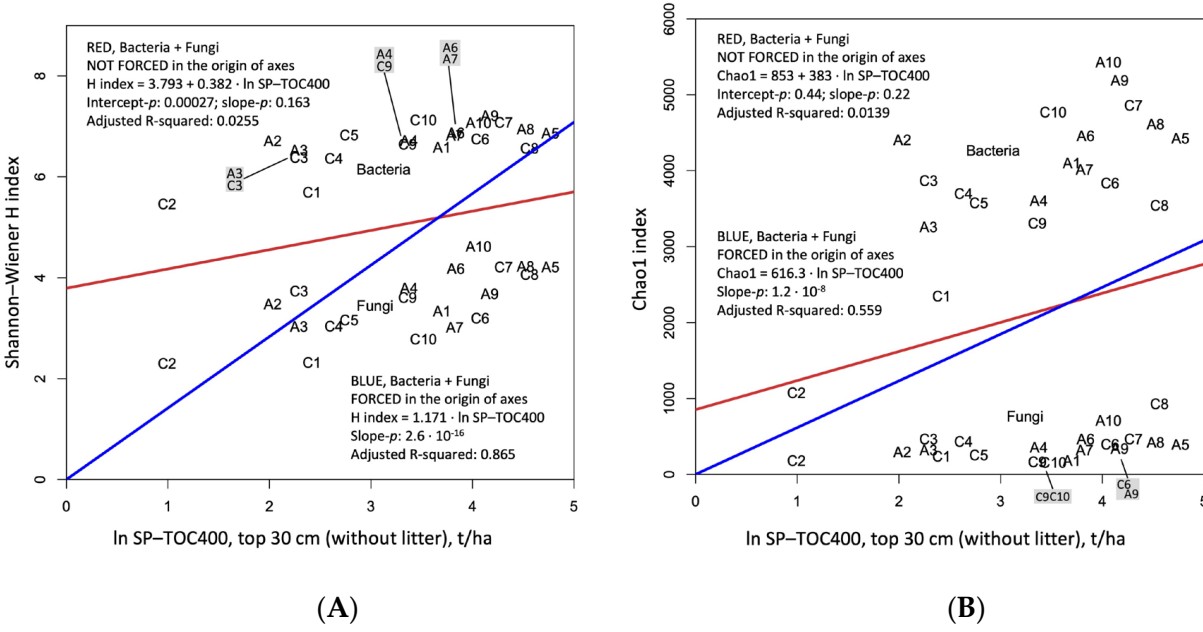

**(A)**                                             **(B)**

**Figure 12.** Linear regressions between (**A**) Shannon–Wiener or (**B**) Chao1 indices (OTUs of bacteria and fungi) and the logarithm (ln) of Skalar Primacs TOC400 (SP–TOC400) measured in Albarella and Caleri.

Unforced in the origin, the regression line even decreased the values of R2, both for Shannon–Wiener and Chao1; forced in the origin, the line showed values that reach above 0.85. Two challenging questions come to mind: (1) Is it right to adopt a logarithmic scale for the variable on the abscissa, thus adding a plateau to the model? (2) Does it make sense to force the model into the origin?

**Answers:**

These are microorganisms, and we know that "they do not know plateau", they grow exponentially until there is no more food. When they run out of provisions, they go into stasis or die. However, in our case, we are talking about growth of a diversity of microorganisms in an ecosystem that lasts over time. It is not the same phenomenon that can be observed in the case of short, localized sources of food. In our case, the system that contains these microorganisms (from young and simple ecosystems near the sea, and along the range to the older and complex woodland far from the sea) allows these communities of

microorganisms to grow their complementarity and to become "microorganisms' systems" within each ecosystem. As demonstrated by Bisschop et al. (2022) [97], the bacterial communities can be shaped by mite ancestry or plant environment; in their experiment, and in just 12 generations of mites, microbiome composition and mite performance were found to be interdependent.

If this makes sense, then the straight line can also be forced into the origin, zero food corresponding to zero diversity, and an increasing diversity responding to a greater availability of food, up to a plateau of resource limit; here, the microorganisms' system is acting at the microscale like a population of macroorganisms would in a macrohabitat. Figure 12A,B would therefore represent the following natural processes:

(1) Expressed by the Shannon and Chao1 indices, biodiversity grows from the left to the right of the graphs, and corresponds to an increasing organic carbon in the first 30 cm of soils;

(2) Bacteria and fungi show the same trend, even if bacterial OTUs return indices almost double those of fungi;

(3) The regression line that best summarizes the plotted points is only the one that is forced into the origin;

(4) To improve the regression, the abscissa axis reports the organic carbon on a logarithmic (ln) scale, and value 1 corresponds to e = 2.718... (about 2.7 t/ha) and 5 to about 150 t/ha of SOC (Figure 10). *Shannon–Wiener* is a logarithmic (ln) index, so the straight line is a good compromise that allows one to understand the growth of biodiversity with the soil carbon content expressed in a logarithmic way. We can say that *Shannon–Wiener* = 1.42 • ln (TOC400).

$$Shannon - Wiener = H_{Shannon-Wiener} = -\sum_{i=1}^{s} (p_i \ln p_i) \tag{1}$$

where *s* is the number of OTUs and *pi* is the proportion of the community represented by OUT *i*.

(5) The *Chao*1 scale is not logarithmic, but the new regression line explains more than 50% of the variance. There is a big difference between fungi and bacteria for this index that gives a lot of importance to the OTUs which are represented by one or two fragments of identification genetic code:

$$Chao1 = S_{Chao1} = S_{obs} + \frac{F1(F1-1)}{2(F2+1)} \tag{2}$$

where *F*1 and *F*2 are the count of singletons and doubletons, respectively, and $S_{obs}$ is the number of observed species.

(6) Less evident on these figures but well-reported in Table 2 and Figure 4, the biodiversity of Albarella is different and higher from that of Caleri. In Figure 12, letters C (Caleri) are shifted down more than letters A (Albarella), for both bacteria and fungi, on average.

*4.2. Soil Biodiversity and Anthropic Soil Use*

Soils are regarded as the habitat harboring the most temporally invariant microbial communities [98] because of the many microhabitats present in soils at very small scales [99], which, in theory, should limit the rate and success of independent dispersal processes [100]. In our study, the stochastic process, especially dispersal limitation, may be most important in regulating the community assembly of soil bacteria in Albarella, while the role of deterministic process, especially variable selection, tended to dominate in Caleri (Figure 8e). Some explanations for this are that habitats with better environmental quality have more individuals, combined with the strong dispersal ability of bacteria, which may lead to an increase in the mass effect [101,102], and the mass effect may contribute to homogeneous selection. Additionally, soil heterogeneity in the natural environment will increase the proportion of variables selected, and the stronger edge effects could also contribute to

the increased role of variable selection [57,103]. Anthropogenic environmental change (i.e., fertilization, mowing, warming) to soils has been shown to mediate the importance of stochastic compared to deterministic processes [43]. In a previous study, stochastic processes increase under relatively high nutrient conditions, while deterministic processes seem to be more related to low-nutrient conditions [80]. However, Ferrenberg et al. (2013) have shown that after the disturbance event, there is a tendency for a stochastic process to evolve into a deterministic process over time [104]. An explanation is that when species arrive through the continued dispersal of the regional species pool, the chance of better-adapted species to arrive at a local site increases [104,105].

In addition, the stochastic process may also have contributed primarily to soil fungal community turnover in both islands, but compared with the Albarella, the undominated process replaced part of homogeneous selection in Caleri (Figure 7e). The increased role of the undominated process might include the effects of a reduced species pool as a result of the disturbance and stochastic colonization of new niche space that led to priority effects [106], which caused communities to diverge from each other in their assembly trajectories, leaving the signature of stochastic community assembly [57].

Overall, different ecological processes may lead to different community assembly of soil bacteria and fungi in natural and anthropic ecosystems.

Anthropogenic environmental change (i.e., fertilization, mowing, warming) to soils has been shown to increase the importance of stochastic compared to deterministic processes [43]. In a previous study, stochastic processes increase under relatively high nutrient conditions, while deterministic processes seem to be more related to low-nutrient conditions [80]. The reasons for such increases in stochasticity in response to environmental changes are still unclear, but might include effects of a reduced species pool as a result of the disturbance and stochastic colonization of new niche space that led to priority effects [105]. Some studies have shown that after the disturbance event, there is a tendency for a stochastic process to evolve into a deterministic process over time. Given the reason that the process of colonizing becomes more important as more species arrive through the continued dispersal of the regional species pool, the better-adaptable species would reach local ecosystems [53,56].

In our case, the low differences in microplastic content between soils of anthropized and natural sites suggest that microplastics are not decisive in the ecological comparison of the two sites (Figure 9). We will deepen the study by focusing on some points that appear to be relatively rich in microplastics (probably the contribution of compost) against others in which they are absent (the most protected from visitors).

We suggest that human activities' disturbances can influence the relative importance of different community assembly processes, and that changes in community assembly processes in response to the human activity's disturbances should be further studied to provide insights into the soil microbial community.

**Biodiversity frontiers**

Working on transects that go from the sea towards the hinterland, we compared an anthropized environment with a more natural one. The vegetation of the 10 ecosystems identified along the transect is characterized by an increasing structural complexity, from the beach colonized by a few species to the innermost multilevel forest. The object of the study is to investigate the microorganisms living in the first 10 cm of soil. The real underlying question remains that of making sense of the word biodiversity. The results just illustrated allow us to conclude as follows:

Comparing equivalent environments from the vegetational point of view, even if the ecosystems with the highest biodiversity (woods) are found in the natural environment, on average, the biodiversity indices are higher in the anthropized environment.

The indices are decidedly higher for bacteria in comparison to fungi both in man-made and natural environments.

We suggest that in anthropized ecosystems, chance is more influential than genetic exchanges on biodiversity. By contrast, in natural environments, selection is a more signifi-

cant factor, and biodiversity in these environments is more phylogenetic. Horizontal gene transfer and illegitimate recombination events could have allowed the microorganisms that reproduce in a more natural environment to respond in a collaborative way to the selection of the environment, consequently expressing a significant phylogeny.

The diversity of these microorganisms (OTUs) is linked to the quantity of organic carbon (TOC400) measured in the first 30 cm of soil with two different methods, Carbon Still Yeomans (650 g) and Skalar Primacs (1 g). The following forced line at the origin explains 85% of the variance: Shannon–Wiener H index = 1.42 • ln (TOC400), ln = natural logarithm.

In our more natural and stable ecosystems of Caleri, microbial diversity was lower than that of the more anthropized ones of Albarella (Figures 4 and 5). The higher the resources (estimated through the measurement of soil organic carbon), the higher the microbial biodiversity when measured along a series ranging from pioneer to forest communities. Few well-organized living beings could make better use of the resources than many living beings not interconnected in their exploitation (Figure 13).

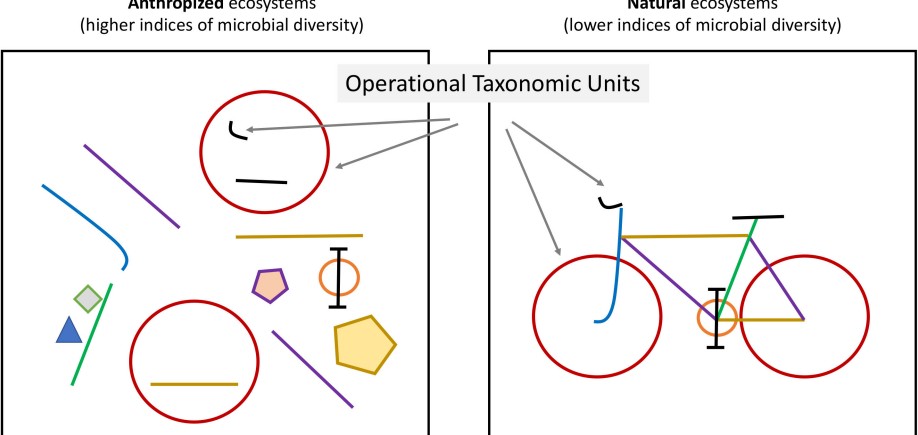

**Figure 13.** Modest honorary gift to John Phillips and Lynn Margulis, with a grateful smile to James Lovelock.

Finally, since its abiotic origin and up to the present day, despite well-known periods of crisis, planet Earth's biodiversity continues to increase. In our work, a delicate phylogenetic relationship was partially revealed by microorganisms cohabiting a more natural environment (in Caleri compared to Albarella). If confirmed (we are still working on it, investigating new soil chemical and biological data), such a cooperating biodiversity [97,107] would acquire a supraorganism functional structure. This paves the way for potential growth over time, on a scale that extends from the smallest to the largest organisms, in the way proposed by Phillips (supraorganism concept), Lovelock, and Margulis (Gaia and Symbiotic Planet, respectively) [2,4]. It would not be competition between living beings that generates evolution, but an intrinsic process of growth of the living system as a whole, in a given and changing environment.

**Author Contributions:** Conceptualization, A.Z., C.B. and A.S.; methodology, A.Z., C.B., L.M., A.S., and software, L.M., M.P., A.S. and A.Z.; validation, A.Z., A.S., I.F., D.B.; formal analysis, A.Z., L.M., A.S. and G.-L.X.; investigation, A.Z., C.B., L.M., A.S., E.L., G.R., M.B., M.C., D.C., M.L. and V.L.; resources, A.Z., C.B., M.R., E.L., L.I. and A.J.Y.; data curation, A.Z., L.M., C.B., A.S., M.P., G.C., G.R.; writing—original draft preparation, A.Z. and L.M.; writing—review and editing, A.Z., L.M., A.S., G.-L.X., D.B., M.P., I.F. and L.I.; visualization, A.Z., L.M.; supervision, A.Z., A.S., G.-L.X., C.B., M.R., E.L.; project administration, A.Z., C.B., M.R., E.L., M.B.; funding acquisition, A.Z., C.B., G.-L.X., D.B., M.R. and E.L. All authors have read and agreed to the published version of the manuscript.

**Funding:** Uni-Impresa project 2020–2022 University of Padua, Italy (EUR 50,000); Associazione Comunione Isola di Albarella, Rovigo, Italy (EUR 50,000); University of Lorraine, France (EUR 14,000); National Natural Science Foundation of China, grant number 42071061, China (EUR 6000).

**Institutional Review Board Statement:** Not applicable.

**Informed Consent Statement:** Not applicable.

**Data Availability Statement:** The BioProject accession number for these SRA data is: PRJNA819224; https://www.ncbi.nlm.nih.gov/sra/PRJNA819224 (accessed on 29 April 2021).

**Acknowledgments:** An affectionate thanks to *Franco Viola*, for his unforgettable lessons on general ecology and population dynamics. A warm thanks also to the *inhabitants of the island of Albarella*, for their collaboration with the university and the logistical support to the student teams.

**Conflicts of Interest:** The authors declare that the research was conducted in the absence of any commercial or financial relationships that could be construed as a potential conflict of interest. Mauro Rosatti is employed by Albarella Srl; Enrico Longo is employed by Associazione Comunione Isola di Albarella. The remaining authors declare that the research was conducted in the absence of any commercial or financial relationships that could be construed as a potential conflict of interest.

**Resource Identification Initiative:** To take part in the Resource Identification Initiative, please use the corresponding catalog number and RRID in your current manuscript. For more information about the project and for steps on how to search for an RRID, please click here.

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
