# Peer review of "Land Use, Microorganisms, and Soil Organic Carbon: Putting the Pieces Together"

_diversity, doi:10.3390/d14080638_

Round 1

Reviewer 1 Report

Presented manuscript „Land use, microorganisms and soil organic carbon: putting the 2 pieces together” is clear and interesting. It could be attractive for scientists working in the field of biodiversity and evolution. The article deals with an current and extremely important issue with is soil microbial diversity change under human pressure. The authors, comparing the ecosystem close to the natural and the second one - transformed by man, identify differences in the microbial diversity of soils in natural and anthropogenic ecosystems, in order to quantify the possible impacts and try to answer questions related to the essence of biodiversity. The results can be the basis for the study of evidence-providing tools to land managers to test and achieve more ecologically-efficient managing practices. Generally the manuscript is scientifically valid, the experimental design is suitable for hypothesis testing, maybe too many hypotheses are tested in one case study.

The methods and results are presented in a clear and accessible way, the illustrative material is well chosen and it is easy to interpret and understand. Figures are appropriate, they properly show the data and are made with great care for aesthetics. The details given in the methods section allow similar experiments to be carried out in other areas. The considerations are supported by the quoted literature, among the publications used there is a significant share of relevant and mostly recent publications (within the last 5 years). There are many older classic articles in the list of references, which is appropriate when authors engage in polemics with well-established definitions. The authors used few references to publications challenging the ideas of Lovelock and Margulis. the number of self-citations is adequate, and the works cited are appropriately related to the topic. The data was interpreted appropriately and consistently throughout the manuscript.

Detailed Notes:

Introduction section is too long. You can shorten fragments about tourism, and information about the location of the trials is included in the methodology, so in the introduction they are partially repeated. The polemics with the definitions of the concept of biodiversity and evolution are controversial, and the data from the two transects also seem to be too little basis for supporting the Gaia concept. I suggest shifting the weight of these considerations from the introduction to the discussion with reference to the results. The results of the experiment are interesting and well described, it seems to me that an approach in which the topic of improving the definition would be abandoned, the whole work would not lose its substantive quality, and would be a more distanced approach to the results.

23-24: specific to “natural plus" 23 or "natural minus" - is not clear

46: “that triggered an Italian-Chinese collaboration" unnecessary

64-67: scientifically irrelevant

Figure 1: 209-215: the legend of the figure is very explanatory, the caption can be shortened to a figure

243: whether it is necessary to give details that the calibration phase was long?

Figure 2: the figure shows the word „shurb” instead „shrub”, the same situation with Figure 4 and Figure 5; instead „wet Land” should be „wetland”?

3.4. Plastics - This chapter seems to be an introduction to another work, isn't it better to make a separate article from the analysis of microplastics? The methodology does not mention in the analyzes, the short paragraph in the rest of the manuscript also adds little to the considerations

607: without „beautiful”

612: no reference to literature

Author Response

Appreciate your suggestions. Please see attached our responses.

Reviewer 2 Report

Dear Authors,

I have reviewed your manusript ''"Land use, microorganisms and SOC: putting the pieces together". Overall, I think this is an interesting piece of work with sufficent data presented.  Below are some moderate comments/suggestions for you to consider:

1) L258-283 can be shorten and wirte in a concise way;

2) Please check carefully all the units or chemical symbol write in a proper way, for instance, CO2 in L260, cm2 in L274, and H2O2 in 279, "2X3" in L344 etc.;

3) check the citations used were fit into the Journal's requirment,  i.e., "(79)-(81)" in L295; "developed by (80)" in L299, "(82), (83)" in L315;

4) check font size throughout the MS, like L341-L344, and others parts.

5) some expressions were unclear in Figure captions, "I level" in L419, “tIOTU” in L422; "BrayCCaleri" in 459;

6) the section (L426-435) was more suitable in INTRODUCTION;

7) I highly suggest to move the subfigures a-d, into a supplentary figure, and combine Fig.7 and 8 together;

8) the subsection 3.4 (L487-495) had a weak link with the context before and after, therefore can be shown as a supplementary file;

9) the letter "A), B)" in figure 10 and 11, should be move onto the very top;

10) L607-621, should be incorporated as INTRODUCTION;

11) L623-L654, this part was not very suitable here, maybe you can pleace part of them in M&M,  and elsewhere.

Author Response

(The authors gave the same response as above.)
